# Empirical Analysis on Public Expenditure for Education and Economic Growth: Evidence from Indonesia

**Agung Suwandaru** [1,*] **, Thamer Alghamdi** [2] **and Nurwanto Nurwanto** [3,4]

1 School of Business, Bankstown Campus, Western Sydney University, Milperra, NSW 2214, Australia
2 School of Business, Parramata City Campus, Western Sydney University, Parramata, NSW 2150, Australia; 19251732@student.westernsydney.edu.au
3 School of Education, Western Sydney University, Penrith, NSW 2751, Australia; 19568847@student.westernsydney.edu.au
4 Department of Islamic Education, Universitas Muhammadiyah Yogyakarta, Bantul 55183, Indonesia; nurwanto@umy.ac.id
* Correspondence: 19587312@student.westernsydney.edu.au

**Abstract:** The purpose of this paper is to find the relationship between public expenditure in the educational sector and the economic growth in Indonesia since the government decided to spend 20% of the state budget on education. We used time series data from 1988 to 2018 and the Cobb–Douglas production function as an economic theory for measurement. In the methodology, we employed Autoregressive Distributed Lag bound tests to find the relationship between variables. The results show that public expenditure on education has an insignificant relationship in the long- and short-term estimation. However, they both have different directions, which is a positive relationship in long-term and a negative relationship in short-term estimation. Meanwhile, gross fixed capital formation shows a positive relationship, and the labour variable has a negative relationship in the short and long terms. In conclusion, the Indonesian government should manage the education system regarding the relationship between education expenditure and economic growth.

**Keywords:** public expenditure; education; economic growth

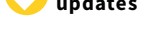

## 1. Introduction

Many countries worldwide have paid attention to education as one of the leading sectors for economic growth. Countries invest in education to elevate their human resources, which will increase growth. The study of Mercan and Sezer (2014) finds that public expenditure on education had a robust impact on Turkey's economic growth from 1970 to 2012. Similarly, using time-series data from 1977–2004, in six countries, namely, Saudi Arabia, Kuwait, Oman, UAE, Bahrain and Qatar, Al-Yousif (2008) demonstrates an effect of education expenses on economic growth. Haini (2020) shows that health and education spending across provinces in China from 1996 until 2015 accelerated economic growth. A meta-analysis study shows that public education expenses positively impact economic growth, while health expenditure negatively affects it (Churchill et al. 2015). Public expenditure in education will compete with public votes for poverty recovery as a short-term program, as occurred in Brazil (Bursztyn 2016). These studies conclude that public expenditure in education is the leading sector for economic growth. Although many countries made a long-term investment in education, it is also a priority as a development program through education.

This study aimed to determine the relationship between education and economic growth in Indonesia's economy. The education sector has been an exciting topic in Indonesia since the government has focused on massive development and increasing a significant expenditure on education. Considering the New Order regime in Indonesia from 1966 until 1998, economic growth had become the top priority. As one of the public sectors, education

had been developed as an arena of flourishing young generations expected to be workers or labourers with specific skills. For instance, the central government issued an elementary school construction policy to carry out its mission. Therefore, education became more accessible. In turn, education is correlated with a higher income (Duflo 2001; Ozturk 2008). This finding can be regarded as a microanalysis, while studies that relate public expenditure on education to economic growth in Indonesia utilising time-series data have rarely been conducted. This paper attempts to demonstrate that educational expenditure coincided with investment and number of labours with economic growth. Within approximately three decades, from 1986–2018, Indonesia's economic figures can be divided into the New Order regime ruling Indonesia until 1997, and the reform era that started in 1998. Globally, the 1998 crisis had a detrimental effect on every household expenditure and public health level (Christian 2002) as it affected every Indonesian household. Therefore, the long-term public expenditure on education and economic returns necessitates further study.

This paper provides a further brief literature review of education and economic growth and explains its data and methodology. Furthermore, the empirical results, discussion and conclusions are presented.

## 2. Literature Review

There has been an ongoing debate on the relationship between education and economic growth. On the one hand, it is argued that education will affect economic improvement, while, on the other hand, it is claimed that any economic condition will affect education. Although we believe that both standpoints can occur simultaneously, we consider education expenditure as an influencing economic growth factor in this paper. However, the impact of public expenditure in education on economic growth remains debatable.

Education has been regarded as an influential force in shaping the economy of any nation. In this sense, education can create either a direct or indirect impact on the economy (Breton 2013). The direct effect means education equips people with knowledge, predispositions, and skills. Thus, education contributions to the economy are complex where education cannot be simplified as labour skills (Lauder et al. 2018). Instead, education also produces individuals who have possible capacities for innovation and the redirection of development. Here, while education has been valued as an arena of boosting human capital (Karaçor et al. 2017), there is an indirect benefit of education for the economy. People with qualifications could generate economic returns. Through studies of over 30 provinces and autonomous regions, directions to optimise human capital in reaching high economic growth in China have flourished (Chang and Shi 2016).

Nonetheless, there has been a challenge for human capital development. For instance, although human capital in Korea remains significant to boost economic growth, the number of elderly or female labours has become another trigger for public policy improvement (Han and Lee 2020). In a sense, Al-Yousif (2008) and Burgess (2016) suggest that aspects of human capital as an educational outcome and the specific policy of each country need to be considered.

Educational institutions have been engineered to produce a high-quality outcome. In comparison, between 1973 and 1978, Indonesia expanded education through elementary schools as basic literacy improvement. Other further education levels, such as higher education, have also been reshaped to fulfil development targets. For instance, families in the East and Southeast Asian countries are expected to graduate from higher education to gain greater access to employment (Postiglione and Wright 2017). In addition, universities have been regarded as a central direction of the knowledge economy since they provide human capital with the skills needed for development (Zaika and Gridin 2020). Another benefit is that education can decrease social injustice. As a local case, expenses in Pakistani higher education have reduced income inequality (Dotti 2019; Qazi et al. 2016).

On the contrary, access to education is also influenced by economic development. For example, data show that economic conditions have impacted the delayed participation

in European universities (Souto-Otero and Whitworth 2017). In other words, education can be improved if economic development itself increases.

Although economic growth can boost education, some countries have had either negative or positive directions. For instance, even though Portugal had experienced a long history of 'counter-cyclical evolution' in which education supported economic growth, economic returns themselves did not fully support the level of education (Nunes 2003). On the other hand, the Finish educational system has been considered one of the leading institutions in building an education-based economy (Schatz 2015). As a result, more attention has been paid to education since a cyclical production between education and economy has been built. Models of education trigger economic benefits and, at the same time, economy emphasizing innovative education remains significant in shaping the future. Despite the positive impacts of education expenditure on economic growth, there is also an urgent need to improve the public spending management system, thereby sustaining development. There should be innovation in education where knowledge is navigated for the economy (Serdyukov 2017).

### 3. Data and Methodology

*3.1. Data*

In this study, we used time-series data to determine the influence of public expenditure on the education sector toward economic growth in the range 1986–2018. The growth of total gross domestic product per capita (Y), labour (L) and gross fixed capital formation (GFCF) data were taken from the World Development Indicators, (World Bank 2020). Data on public expenditure on education were obtained from Statistik Indonesia (Indonesia Statistik 2020), an agency within the Indonesian government responsible for conducting statistical surveys. All data were in the form of real terms and were changed to the form of natural logarithms.

The endogenous growth model (Cozzi 2017; Gualdi and Mandel 2019) holds that economic growth is influenced by public capital investment. Consequently, public capital investment will be the major component in shaping economic growth. Public expenditure on education is a part of public capital investment, so the increased allocation for the education sector will impact economic growth.

This study was based on the Cobb–Douglas production function with constant return technology. This function measures the magnitude of several input variables' relationships to the number of production factors (Dritsaki and Stamatiou 2018). We assumed that increasing human capital positively influences a country's economic growth. Thus, the formula for the Cobb–Douglas production function is as follows:

$$Y_{nt} = A_t K_t^{\mu} L_t^{\alpha-1} \tag{1}$$

*Y* is the dependent variable; it indicates the growth of the total gross domestic product per capita, whereas *A* shows the level of technology used and experiences a constant increase even in overtime conditions. We postulated that an increase in physical capital accumulation does not affect the level of technology used. *K* is the physical capital stock, and *L* indicates the number of workers.

We assumed that economic growth is affected by investment and fixed physical capital. This idea is supported by Boamah et al. (2018); Kharusi and Ada (2018), in different implementations. Therefore, we conceptualise physical capital stock as public expenditure on education and physical capital sectors.

$$K = f\left(Edu_t, \ GFCF_t\right) \tag{2}$$

Then, we have

$$Y_t = A_t Edu_t^{\mu 1} GFCF_t^{\mu 2} L_t^{1-\mu} \tag{3}$$

We converted the equation into multivariate regression in natural logarithms:

$$lnY_t = \alpha_t + \mu_1 \, ln \, EDU_t + \mu_2 \, lnGFCF_t + (1 - \mu)lnL_t \tag{4}$$

To minimise the bias heteroscedasticity problem in the regression model, we changed variables into natural logarithms. We also used a series of unit test root tests on time series data to use our model to observe the stationarity of the data. Stationary data comprise a data variant that undergoes systematic changes over time. These data can avoid heteroscedasticity and autocorrelation problems in a model. We applied Augmented Dickey–Fuller (ADF) and Philip Perron (PP) unit root tests to measure the stationarity of the data. The regression model used for the ADF test is based on the following equation:

$$\Delta \, Y_t = \alpha + \beta Y_{t-1} + \sum_{i=1}^{k} \pi_i \Delta Y_{t-1} + \varepsilon_t \tag{5}$$

where $Y_t$ is a variable with time $t$, $\Delta$ is the first difference of variable, $\alpha$ indicates a constant, and $\varepsilon$ is the error term. According to Paparoditis and Politis (2018), a stationary variable has a negative value that is greater than the critical value that has been determined.

We also applied the Philip Perron unit root test. This test has no assumptions on the residual value and considers the error value to be normal. Moreover, the Philip–Perron test is more consistent on heteroscedasticity than the Dicky–Fuller statistical test. Technically, the Philip–Perron test proposes that stationary data have a t-statistic value higher than the predetermined critical value. Furthermore, the *p*-value is less than 0.05, so the data series is stationary. The Philip–Perron test was built with the following equation:

$$\Delta Y_t = \gamma + \delta Y_{t-1} + \varepsilon_t \tag{6}$$

### 3.2. The Auto-Regressive Distributed Lag

According to Nkoro and Uko (2016), a regression model that includes variable values that explain the present value or the past value (lag) of the dependent variable as one of the explanatory variables is called auto-regressive distributed lag (ARDL). We used the ARDL model to analyse the effect of public expenditure on the education sector on economic growth. The equation model is as follows:

$$\Delta \, Y_t = \rho_0 + \alpha_1 lnY_{t-1} + \alpha_2 lnEDU_{t-1} + \alpha_3 lnGFCF_{t-1} + \alpha_4 lnL_{t-1} + \sum_{i=1}^{p} \delta_i \Delta lnY_{t-i} + \sum_{i=1}^{p} \vartheta_i \Delta lnEDU_{t-i}$$
$$+ \sum_{i=1}^{p} \gamma_i \Delta lnL_{t-i} + \varepsilon_t \tag{7}$$

The ARDL model can estimate short- and long-term relationships between variables (Kripfganz and Schneider 2016). This model is also very consistent with a small sample size. Furthermore, ARDL can correct residual and endogenous variable problems simultaneously.

### 3.3. Wald Cointegration Test

The Wald test shows that there is a long-term coefficient of information in the variable. According to Nathan and Liew (2013), the Wald test value can be obtained by comparing the F-statistic value with the critical value limit at the lower I (0) and upper I (1) bounds. The Wald test's null hypothesis is that the F-statistics are lower than the values of upper and lower critical bounds. An inclusive decision is made on a variable if the value is between the lower and upper critical bounds.

$$y_t = \rho_1 + \sum_{i=1}^{l+m} \delta_{1i} y_{t-i} + \sum_{i=1}^{l+m} \delta_{2i} x_{t-i} + \varepsilon_{yt} \tag{8}$$

$$x_t = \rho_2 + \sum_{i=1}^{l+m} \alpha_{1i} y_{t-i} + \sum_{i=1}^{l+m} \alpha_{2i} y_{t-i} + \varepsilon_{xt} \tag{9}$$

$\varepsilon$ is the white noise error terms, whereas *l* is the optimal lag of the model, and *m* is the maximum integration value. The Wald test value can be found with the Chi-squared value.

### 3.4. Vector Error Correction Model (VECM)

The ARDL model calculation will usually be diagnosed with significant results by including long- and short-term lag. The long-term lag can use the following equation:

$$Y_t = \gamma_0 + \gamma_1 \sum_{i=1}^{a} Y_{t-1} + \gamma_2 \sum_{i=1}^{a} lnEDU_{t-1} + \gamma_3 \sum_{i=1}^{a} lnGFCF_{t-1} + \gamma_1 \sum_{i=1}^{a} lnL_{t-1} + \partial_t \quad (10)$$

If there is a long-term lag in the diagnosis, then the short-term lag can be found with the following equation:

$$\Delta Y_t = \gamma_0 + \gamma_1 \sum_{i=1}^{a} Y_{t-1} + \gamma_2 \sum_{i=1}^{a} lnEDU_{t-1} + \gamma_3 \sum_{i=1}^{a} lnGFCF_{t-1} + \gamma_1 \sum_{i=1}^{a} lnL_{t-1} + nECT_{t-1} + \partial_t \quad (11)$$

The error correction term (*ECT*) indicates the speed of adjustment and shows how quickly the variable returns to long-term equilibrium. The *ECT* should be between zero and one in the negative form to indicate a long-term relationship.

### 3.5. Stability Check

In this study, several tests were used to check the robustness of the model. First, the Jarque–Bera test was used to determine whether residual values were normally distributed or not (Kim 2016). Second, we employed the White test to identify the factors that cause the model to be inefficient and inaccurate (Hausman and Palmer 2012). Third, the Breusch–Godfrey LM test detected serial correlation problems. Fourth, the Durbin–Watson test was applied for autocorrelation regression measurement in residuals (Bercu et al. 2015). Finally, we used the cumulative sum on the recursive residuals (CUSUM) to explain shifts in mean or variance, and CUSUM of square (CUSUMSQ) tests detected rapid changes from a constant regression coefficient (Riaz et al. 2011).

### 3.6. Analysis Variables

Indonesia's average economic growth in the period 1986 to 1997 was almost 7%. This range period has made Indonesia able to improve from a low-income country into a lower-middle-income country category. The growth trend from 1986 to 2018 is shown in Figure 1, gradually increasing, except in year 1998, which declined significantly due to the financial crisis. This crisis affected all sectors, especially the economic sector. After the year 2000, the Indonesian economy experienced a recovery with an average growth of around 4.6% per year. This economic recovery was primarily supported by rising household consumption and the explosion of commodity prices in the 2000s.

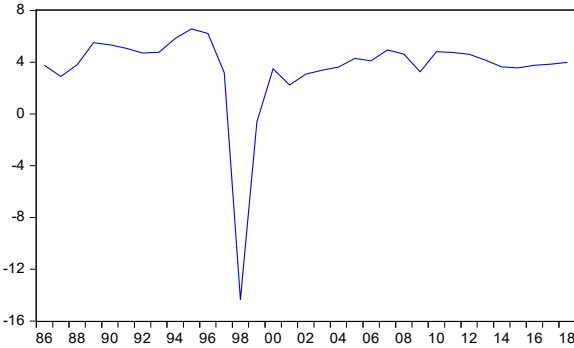

**Figure 1.** Trends of growth of GDP in Indonesia, 1986–2018.

Education access in Indonesia is essential to improve the quality of human resource development in the future and to bridge the education gap from other countries. By in-

creasing the education expenditure, education access becomes inclusive for the population. Sequentially, the quality of human resources will be improved to become globally competitive. The education policy is in line with the mandate of the 1945 Indonesia Constitution, article 31; in paragraph (1), each citizen has the right to an education; in paragraph (2), each citizen in obliged to follow elementary education, and the government has to fund this; in paragraph (3), the state gives priority to the education budget by the allocation of at least 20% of the state as well as the regional budgets to meet the requirements of implementing national education.

Furthermore, the budgeting for education funding is strengthened by Law number 20 of 2003, which concerns the national education system (Sistem Pendidikan Nasional/Sisdiknas). This law in article 49, paragraph 1, states that education funds other than teacher salaries and official education costs are allocated a minimum of 20% of the State Budget (APBN) for the education sector. The same percentage is also mandated to be allocated by each region to their respective regional budget (APBD).

Although the allocation of funds is clearly stated, this policy does not mean that it can be implemented directly. Political tug-of-war, debates in the mass media, and lawsuits in the constitutional court (MK) during the period 2005–2008 prevented the implementation of the law, and finally, the education funds could be applied in 2009. At that time, the 2009 education budget was allocated a drastic increase of 20% of the state budget or IDR 207.4 trillion. From 2009 to 2014, the education budget allocation was the second-largest portion of the APBN spending after subsidy spending, whereas since 2015, the education budget has become the largest government expenditure. Figure 2 shows a significant increase in 2009 compared to the other years. A slight decline in the budget occurred in 1998 due to the financial crisis.

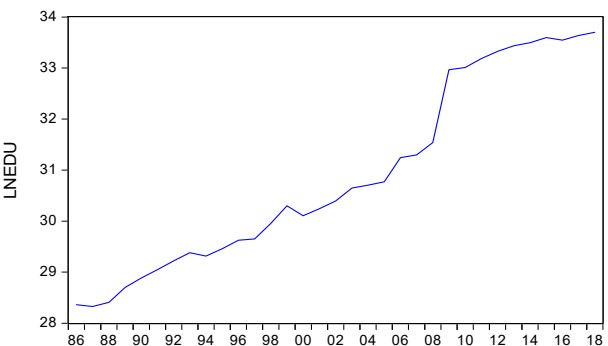

**Figure 2.** Trends in public expenditure on education in Indonesia, 1986–2018.

GFCF, or so-called investment, is the number of assets acquired for production activities over a certain period less the asset's residual value. In this study, GFCF was considered as a capital that can affect GDP in Indonesia. This assumption is relevant to Chow (2017); Gibescu (2010), who stated that GFCF influences changes in GDP. The GFCF trend in Figure 3 shows a sharp increase after 1987 and a drastic decline in 1997 due to the financial crisis that hit Asia. The GFCF recovered moderately between 1999 and 2000. In 2001, the GFCF trend experienced a slight decline due to the uncertain domestic political situation, the impact of the dot-com bubble collapsed and the recession in the US. After the year 2002, GFCF experienced a significant increase as economic development had begun to improve.

The trend of the increasing labour force in Indonesia is a logical consequence of the population growth every year. Indonesia is a country that has a relatively large population, especially in relation to the labour force. The average growth in the labour force in Figure 4 shows an increase of approximately 1–3% per year. However, Figure 4 shows an increasing labour force by 5.58% in 1994 and a decrease of below 1% in 2006, 2011 and 2015.

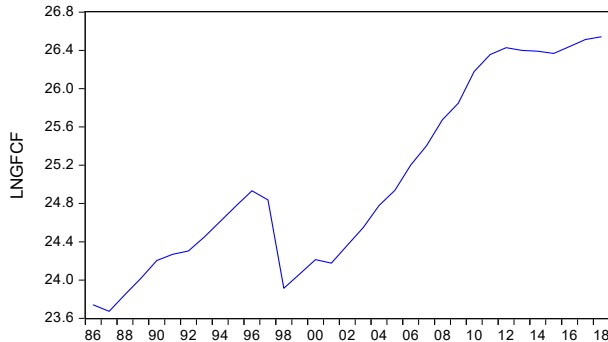

**Figure 3.** Trends of GFCF in Indonesia, 1986–2018.

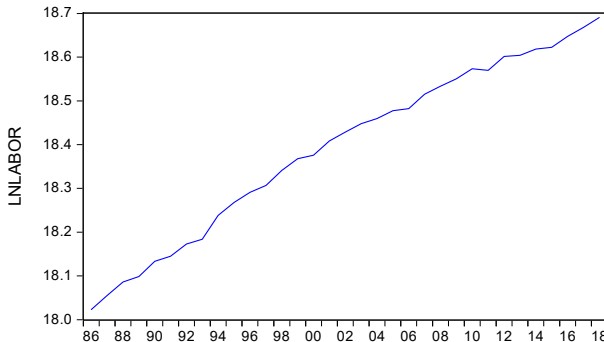

**Figure 4.** Trends of labour in Indonesia, 1986–2018.

## 4. Results

### 4.1. Unit Root Test

We implemented the ADF test with selected lag lengths using the Schwarz Bayesian Information Criterion (SBIC) model (Chakrabarti and Ghosh 2011). We used the procedure based on Narayan and Popp (2010), which allows a single structural break for a unit root test. Table 1 shows that all variables, except for lnY variable, are not stationary at the level of the SBIC test. We repeated the test on the first difference for the lnEDU, lnGFCF and lnL variables that were not stationary at level, then revealed the variables that were stationary. Therefore, all confirmed variables were highly suitable for the application of the ARDL bounds test method. Furthermore, we applied the Philip–Perron (PP) unit root test to compare the null hypothesis, which was not stationary, to the alternative hypothesis, which was stationary. The tests' diagnosis results in Table 1 show that mostly the variables were not stationary at level, but they were stationary at the first difference. Moreover, the lnY variable was the only variable which was stationary at level.

**Table 1.** Stationary test results.

| Variables | ADF (SIC) Linear Trend | | Philip–Perron Linear Trend | |
|---|---|---|---|---|
| | Level | First Difference | Level | First Difference |
| lnY | −4.56 *** | ∅ | −4.55 *** | ∅ |
| lnEDU | −2.06 | −5.63 *** | −2.17 | −5.63 *** |
| lnGFCF | −1.52 | −4.34 *** | −1.75 | −4.35 *** |
| lnL | −1.01 | −8.21 *** | −1.34 | −9.38 *** |

Note: *** represents 1% level of significance, respectively. ∅ sign indicates that the variable is stationary at level.

### 4.2. Cointegration Tests

Based on the Wald test diagnosis results, Table 2 shows that the F-statistic value is 43.79, which is greater than the upper and lower bound statistic value above a critical value

of 1%. According to Cho et al. (2015), if the F-statistic value is greater than the critical value, then there is a long-term relationship or cointegration between the GDP and public expenditure on education GFCF and labour. Thus, the results of the cointegration indicate that all variables have a long-term equilibrium.

**Table 2.** Wald's cointegration test.

| Wald F-Statistic | 10% | | 5% | | 1% | |
|---|---|---|---|---|---|---|
| | Upper Bound l(1) | Lower Bound l(0) | Upper Bound l(1) | Lower Bound l(0) | Upper Bound l(1) | Lower Bound l(0) |
| 43.79 | 2.72 | 3.77 | 3.23 | 4.35 | 4.29 | 5.61 |

Table 3 shows the error correction term (ECT) test results for the selected ARDL (1, 2, 1, 0). The probability value on ECT is 0.000, indicating that it is significant. Based on the ECT value, we conclude that Equation (11) is a valid model for short-term dynamics in this study. The ECT coefficient shows the speed of adjustment from short- to long-term equilibrium of −0.924954 or 0.0092% of the model distortion that occurred in the first year. Moreover, all variables show significant probability values, except for the education variable in lag 1. Finally, we can confirm that the variable will have a significant impact on the growth depending on the duration, which is shown in the lags of the model.

**Table 3.** Short-term coefficients estimations.

| Variables | Coefficients | t-Statistics | Probability |
|---|---|---|---|
| D(lnEDU) | −0.108010 | −1.587780 | 0.1260 |
| D(lnEDU)(−1) | −0.171526 ** | −2.723074 | 0.0121 |
| D(lnGFCF) | 1.140276 *** | 13.629019 | 0.0000 |
| D(lnL) | −0.643257 * | −2.032553 | 0.0538 |
| ECT(−1) | −0.924954 *** | −11.546337 | 0.0000 |
| ARDL (1, 2, 1, 0) | | | |

Note: ***, **, * represents 1%, 5%, and 10% levels of significance, respectively.

The test result in Table 4 shows that the impact of public expenditure on the education sector on economic growth is a weak positive relationship in the long term. The constant value 12.365414 indicates an increase of 12.365% annually if there is no annual growth of public expenditure on education, GFCF and labour. The long-term coefficient value of 0.028864 for variable public expenditure in the education sector shows that an increase of 1% will increase economic growth by 0.00029%. Supportive research in line with the result was conducted by Bosupeng (2015). However, the findings of several studies show contrasting results to those of this study. Their research results show that public expenditure on the education sector positively and significantly impacts economic growth (Baldacci et al. 2008; Blankenau et al. 2007; Ifa and Guetat 2018; Kushwaha and Tiwari 2019). A study with negative results for the impact of public expenditure on education sector on economic growth was conducted by Keller (2006). Moreover, the estimation results indicate that public expenditure has a negative but insignificant impact on education in the short-term estimation findings of Ifa and Guetat (2018). In contrast, Rahman (2011) and Kushwaha and Tiwari (2019) found that the effect of education expenditure on economic growth is positive and significant in the short term.

The GFCF variable shows a significant influence on economic growth in the long and short term. The positive coefficient value indicates the positive effect of GFCF on economic growth with a value of 0.106467 for the long term and 1.140276 for the short term. In other words, a 1% increase in GFCF will increase the economic growth by 0.0011% in the long term and 0.011% in the short term. This finding is in line with the findings of Gibescu (2010); Sharma and Mittal (2019).

**Table 4.** Long-term coefficients estimations.

| Variables | Coefficients | t-Statistics | Probability |
|---|---|---|---|
| lnEDU | 0.028864 | 0.445045 | 0.6604 |
| lnGFCF | 0.106467 | 1.385521 | 0.1792 |
| lnL | −0.695447 * | −1.964918 | 0.0616 |
| C | 12.365414 ** | −2.116395 | 0.0453 |

Note: **, * represent 5%, 10% levels of significance, respectively.

Labour is an essential variable in shaping long- and short-term economic growth. The coefficient elasticity value in this study was not significant in either estimation. These significant long- and short-term effects have elasticity levels of −0.695447 and −0.643257. This means that a 1% increase in labour led to a decrease in the economic growth of 0.0069% in the long term and 0.0064% in the short term, respectively. The results confirm that the labour force has different relationships with economic growth Auzina-Emsina (2014); Brancaccio et al. (2018). In contrast to the findings of Amaral (2009), Herwartz and Niebuhr (2011) and Ismail and Yuliyusman (2014), their study state that the labour variable has a positive and significant relationship with economic growth.

### 4.3. Stability Checks

The stability of the model in this study is robust and can be seen in Table 5. The Jarque–Bera test shows that the probability value of 0.9977 is greater than 0.05. This means that the data are typically distributed. Meanwhile, White's test with a probability value of 0.2494 indicates that we cannot reject the null hypothesis of homoscedasticity between variables.

**Table 5.** ARDL diagnostic tests.

| Stability Check | | Variables | |
|---|---|---|---|
| **Test** | **Purpose of Tests** | ***p*-Value** | **Decision** |
| Jarque–Bera test | Normality Test | 0.9977 | Normal |
| White's test | Heteroscedasticity Test | 0.2494 | No heteroscedasticity |
| Breusch-Godfrey LM test | Serial Correlation Test | 0.4095 | No serial Correlation |
| Durbin-Watson | Autocorrelation Test | 2.1845 | No autocorrelation |

The Breusch–Godfrey LM test detects autocorrelation in variables. The test result obtained a probability value of 0.4095, which is greater than 0.05. It can be concluded that there is no autocorrelation between variables. To check the stability of the model, CUSUM was used, as shown in Figures 5 and 6. The model was shown to be consistent with predetermined parameters, and there is no deviation throughout the sample periods.

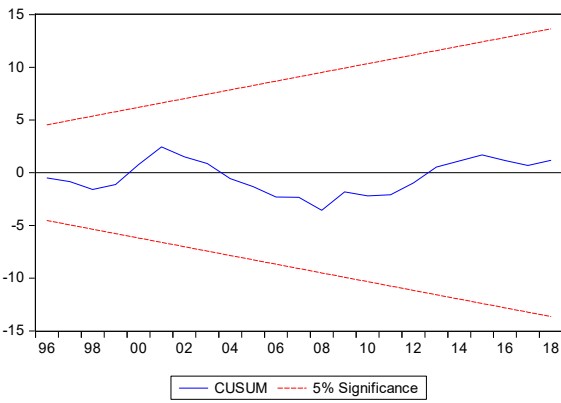

**Figure 5.** CUSUM test.

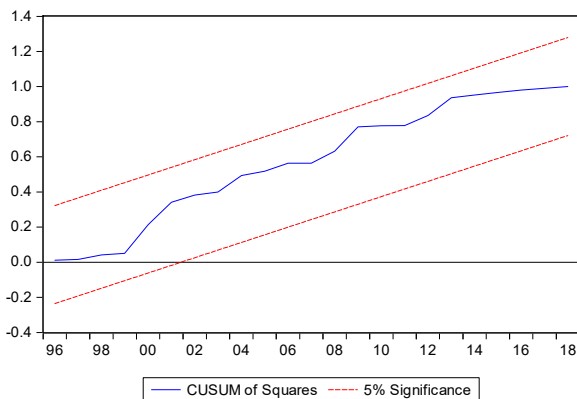

**Figure 6.** CUSUM squared.

## 5. Discussion

The Indonesian government's enthusiasm to improve the economy by investing in education should be appreciated as implementing its constitution. The constitution mandates allocating 20% of the APBN of the Indonesian government to improve its economy via education investment. From this study, we can see the increase in public expenditure in education starting in 2009. There were many obstacles, mostly in politics and bureaucracy, to implementing the mandate of law number 20 of 2003 concerning Sisdiknas, until it was implemented in 2009. We can see the drastic increase in public expenditure in education starting in that year. Regarding the education spending, this study attempted to find the relationship between spending and economic growth. The results show that there is a significant relationship in the long-term estimation. This means that the Indonesian policy on education expenditure is on the right track. However, the transmission of public education expenditure is relatively slow. Our results support those of Bosupeng (2015) and Wardhani et al. (2017), showing that public expenditure is insignificant to economy performance.

We also mirror the results of law number 20 from 2003 that was implemented in 2009. Indonesia is considered as a corrupt country (Svensson 2005), including their public expenditure in education (Kristiansen and Santoso 2006). The studies of Olken (2006, 2007) demonstrate that corruption in the government contributes to bureaucracy effectiveness in Indonesia. The gaps of the implementation of a national education system could be considered less effective in the bureaucracy between 2003 and 2009.

Furthermore, education expenditure contributed a weak positive effect to the economic growth. We consider the study of Kristiansen (2006), which found that the decentralization law in Indonesia has no discernible effect on economic performance, which is in line with the results of this study. Decentralization concepts allow local governments in Indonesia to manage their own budget. On the other hand, decentralization creates many posts in bureaucracy, which leads to lower efficiency and making corruption easier for state officials. Less effective governance and corruption contribute to absorbing the state budget, which has a negative effect on the economy. The state budget on the education sector, one of greatest proportions in the APBN, gains a large amount and many attentions in the government to improve human development. However, the education public spending has a negligible effect on the economy (Suryadarma 2008).

## 6. Conclusions

This study aimed to determine the relationship between public expenditure on education and economic growth with time-series data for 32 years (1986–2018) in Indonesia. We adopted the endogenous growth theory framework and used the long- and short-term ARDL considering small data. This study showed the contradictory empirical results between long- and short-term estimates of the relationship between public expenditure on education and Indonesia's economic growth. We found a positive and insignificant long-term relationship between public expenditure on education and economic growth. This

result indicated that the increase in education budget allocation was not accompanied by the change in policies that support economic growth. An increase in education expenditure would have a positive relationship and be insignificant for economic growth. Indonesia should change the long-winded bureaucracy culture and administrative bureaucracies, which are the factors inhibiting the success of expenditure for education to the growth. Consequently, even though the decentralization and corruption are matters that still need to be solved, creating an efficient and effective bureaucracy and fighting against corruption should be supported by the government.

In contrast, the short-term relationship between public expenditure on the education sector and economic growth was negative. We highlight that corrupt bureaucracy makes economic transmission ineffective. The public spending is absorbed by the corrupt bureaucracy system due to their additional revenue; hence, the purpose of public spending is not fulfilled. As an alternative solution, an efficient, transparent and effective bureaucracy in the education sector should be implemented in Indonesia so that public expenditure will fulfil its purpose. The findings in this study can be used as an example in countries with similar economic characteristics for further research.

**Author Contributions:** Conceptualisation, A.S.; methodology, A.S.; software, T.A.; validation, A.S., T.A. and N.N.; formal analysis, A.S. and N.N.; writing—original draft preparation, A.S. and N.N.; writing—review and editing, A.S. and N.N.; visualisation, T.A. All authors have read and agreed to the published version of the manuscript.

**Funding:** This research received no external funding.

**Informed Consent Statement:** Not applicable.

**Data Availability Statement:** Not applicable.

**Acknowledgments:** We thank the Ministry of Religious Affairs (MoRA), Republic of Indonesia scholarship and Ministry of Finance, Saudi Arabia scholarship.

**Conflicts of Interest:** None of the authors had a conflict of interest.

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
