# Peer review of "Empirical Analysis on Public Expenditure for Education and Economic Growth: Evidence from Indonesia"

_economies, doi:10.3390/economies9040146_

Round 1
Reviewer 1 Report
The research is very significant and applicable to other educational politics because the purpose of this paper is to find a balance
between public spending on the education sector and economic growth in Indonesia since when the government decided to spend 20% of the state budget on education.
The findings in this study
can be used as an example in countries with similar economic characteristics.
Reviewer 2 Report
Dear authors,
The idea of the theme of this article is interesting and pertinent but the article presents the need for several improvements:
1. The article's abstract is somewhat confusing and contradictory and should be improved to make it more understandable;
2. The literature review presents very old works and should be updated;
3. The article ends with the "Discussion" section. This discussion does not cite any work, which leads me to conclude that it is not a discussion but a conclusion that can and should be improved;
4. I question why the "Conclusions" are not presented;
In short: the article should be greatly improved before it can be published.
Best Regards
Reviewer 3 Report
I believe the article has a strong potential, but it needs some serious corrections. My most importants reservations are following:
- I would actually recommend focusing on per capita GDP growth, rather than on the total GDP growth.
- Why would the authors include the public expenditure on education multiplied by the gross fixed capital formation in the Cobb-Douglass production function? This does not make much sense, does it?
- Since it is very obvious from the explanation of authors and also from the fig. 2, that there was a major breakpoint in government expenditure on education in 2009, why is it necessary to carry out the structural breakpoint analysis? Only to confirm it was a major breakpoint? I would suggest restructuring the article.
- The value of constant C in tab. 3 is rather large and negative. That would actually mean, with no annual growth of public expenditure on education, no annual growth of gross fixed capital formation and no annual growth of labour the real GDP would decline by 12 % annually. Why would that be so? Because of investment depreciation? Or what would be the explanation for that steep economic decline?
- There is an error in an order of magnitude in the sentence: "The long-term coefficient value of 0.030521 for variable public expenditure in education sector shows that an increase of 1% will increase economic growth by 3.0521%." In fact, it would be only by 0.0030521%.
Round 2
Reviewer 2 Report
Dear authors,
After read this second version of the paper I am satisfied with the authors work.
Congratlations